# Transcriptional analysis of amino acid, metal ion, vitamin and carbohydrate uptake in butanol-producing *Clostridium beijerinckii* NRRL B-598

**Maryna Vasylkivska**[1]*, **Katerina Jureckova**[2], **Barbora Branska**[1], **Karel Sedlar**[2], **Jan Kolek**[1], **Ivo Provaznik**[2], **Petra Patakova**[1]

1 Department of Biotechnology, University of Chemistry and Technology Prague, Prague, Czech Republic,
2 Department of Biomedical Engineering, Brno University of Technology, Brno, Czech Republic

* vasylkim@vscht.com

**Data Availability Statement:** The genome assembly referred to in this paper is version CP011966.3, available in the GenBank database.

## Abstract

In-depth knowledge of cell metabolism and nutrient uptake mechanisms can lead to the development of a tool for improving acetone-butanol-ethanol (ABE) fermentation performance and help to overcome bottlenecks in the process, such as the high cost of substrates and low production rates. Over 300 genes potentially encoding transport of amino acids, metal ions, vitamins and carbohydrates were identified in the genome of the butanol-producing strain *Clostridium beijerinckii* NRRL B-598, based on similarity searches in protein function databases. Transcriptomic data of the genes were obtained during ABE fermentation by RNA-Seq experiments and covered acidogenesis, solventogenesis and sporulation. The physiological roles of the selected 81 actively expressed transport genes were established on the basis of their expression profiles at particular stages of ABE fermentation. This article describes how genes encoding the uptake of glucose, iron, riboflavin, glutamine, methionine and other nutrients take part in growth, production and stress responses of *C. beijerinckii* NRRL B-598. These data increase our knowledge of transport mechanisms in solventogenic *Clostridium* and may be used in the selection of individual genes for further research.

## Introduction

Until the 1950s, ABE fermentation by clostridial species was one of the largest industrial biotechnological processes, the second largest after ethanol fermentation. It became unprofitable after the development of petrochemical methods for solvent production, methods which have been used exclusively until now. However, due to environmental concerns and resource limitations there is a demand for renewable methods of fuel and chemical production, independent of crude-oil [1]; ABE fermentation as an alternative and ecological process is now being reinvestigated [2].

The major products of ABE fermentation are butanol, acetone and ethanol, which in the well-studied solventogenic strain *Clostridium acetobutylicum* ATCC 824, are usually produced

RNA-Seq data are available from the NCBI Sequence Read Archive (SRA) under the accession number SRP033480.

**Funding:** This work was supported by project grant GACR 17-00551S, The Czech Science Foundation GACR (https://gacr.cz) to PP. The funders had no role in study design, data collection and analysis, decision to publish, or preparation of the manuscript.

**Competing interests:** The authors have declared that no competing interests exist.

at a molar ratio of 6:3:1 [3]. Butanol has two main applications–it can be used in the fuel industry as a biofuel or an addition to gasoline, and in the chemical industry. It is compatible with the original gasoline engines without modifications [4] and is listed as a preferable green solvent in solvent selection guides [5,6].

Major problems of ABE fermentation are the high cost of raw materials and low final solvent titers. While some clostridial strains are able to utilize a wide range of carbohydrates other than glucose, they still need growth factors such as vitamins and trace elements for growth and metabolite production. Many researchers have reported that addition of appropriate concentrations of amino acids and metal ions to the cultivation medium stimulate bacterial growth and result in increased solvent yield [7–10]. Therefore, an in-depth understanding of transport and metabolism of amino acids, sugars, vitamins and metal ions is important for the selection of proper and inexpensive substrates for ABE fermentation and can potentially result in a decreased solvent price [11,12].

Issues of commercialization of ABE fermentation can be addressed by the use of waste materials as a substrate, application of different fermentation techniques, optimization of fermentation conditions or strain engineering with a focus on flux redirection. All these approaches require an in-depth understanding of metabolic traits and uptake mechanisms.

The uptake of different substances in bacteria is carried out via different types of membrane transporters (Transporter Classification Database http://www.tcdb.org/), including: electrochemical potential-driven transporters (for example, symporters/antiporters transporting metal ions and amino acids), primary active transporters (for example, ATP-binding cassette transporters co-called ABC transporters, including Energy Coupling Factor-Type ABC Transporters for vitamin uptake, and P-type ATPases carrying metal ions) and group translocators (for example, phosphoenolpyruvate-dependent phosphotransferase system PTS for sugar uptake). While a few research papers describing mechanisms of carbohydrate uptake in *Clostridium* exist [12–14], transport of amino acids, vitamins and metal ions have received little attention, despite their major influence on growth and production [15].

*C. beijerinckii* NRRL B-598, formerly known as *Clostridium pasteurianum* NRRL B-598 [16], used in this study, is a robust solventogenic strain with stable production rates and, despite being anaerobic, tolerant to short exposures to oxygen. The whole genome sequence [17], as well as transformation techniques [18] are available for this strain, which therefore has potential for industrial use. However, modification and optimization of cultivation conditions and the fermentation medium are still required. In this paper, we identify genes encoding putative transporters of amino acids, metal ions, vitamins and carbohydrates in *C. beijerinckii* NRRL B-598, combined with their transcriptomic data obtained from the whole life cycle of the bacterium, including acidogenesis, solventogenesis and sporulation.

## Material and methods

### Cultivation experiment

*C. beijerinckii* NRRL B-598 was used in this work. The strain was maintained in distilled water in the form of a spore preserve at 4˚C. Prior to inoculation, the spore preserve was heated to 80˚C for 2 min.

An inoculum for fermentation experiments was prepared in Erlenmeyer flasks containing TYA medium (20 g/L glucose, 2 g/L yeast extract (Merck), 6 g/L tryptone (Sigma Aldrich), 0.5 g/L potassium dihydrogenphosphate, 3 g/L ammonium acetate, 0.3 g/L magnesium sulfate heptahydrate, 0.01 g/L ferrous sulfate heptahydrate) and cultivated at 37˚C overnight in anaerobic chamber (Concept 400, Ruskinn Technology) under a stable $N_2/ H_2$ (9:1) atmosphere.

Cultivation experiments were performed in parallel Multiforce 2 bioreactors (Infors HT, Switzerland) containing TYA medium with 50 g/L glucose. Prior to inoculation, the atmosphere in the bioreactors was switched to anaerobic by bubbling with $N_2$ for 30 min and the inoculum was checked under the microscope. Bioreactors were inoculated with a 10% v/v overnight culture. Before cultivation, the pH was adjusted to 6.3 using 10% sodium hydroxide solution, and during cultivation, the pH was measured but not controlled.

## Analytical methods

Cell growth was determined by measuring the optical density (OD) of the culture broth at 600 nm on a Varian Cary® 50 UV-Vis spectrophotometer (Agilent). TYA medium was used as a blank, and a 10 mm cuvette containing 1 ml of culture broth was used for analysis. Samples taken after the 4th hour of cultivation were diluted so they were within $OD_{600} = 0.1–1.0$ and results were multiplied by the dilution factor.

HPLC with refractive index detection (Agilent Series 1200 HPLC) equipped with an 250x8 mm Polymer IEX H form 8 um column (Watrex) was used for the determination of glucose consumption and production of acids and solvents. Before analysis, samples of culture broths were centrifuged and filtered through 0.2-μm syringe filters. A mobile phase of 5 mM $H_2SO_4$ was used in the experiment. Conditions of analysis were those as described in Kolek et al. (2016): isocratic elution, stable flow rate of 0.5 mL/min, stable column temperature of 60˚C, sample volume for the injection 20 μl [19].

## RNA sequencing

Culture broth samples for RNA isolation were taken at 3.5, 6, 8.5, 13, 18 and 23 h of cultivation (further referred to as sampling points T1, T2, T3, T4, T5 and T6 respectively). Each broth sample was centrifuged, the cell pellet was washed with sterile distilled water and stored immediately at– 70˚C. These sampling points were chosen so that transcriptome data would cover the whole life cycle of *C. beijerinckii* NRRL B-598 [20].

Commercially available kits were used for total RNA isolation (High Pure RNA Isolation Kit, Roche) and ribosomal RNA depletion (The MICROBExpress™ Bacterial mRNA Enrichment Kit, Ambion). Prior to library construction and sequencing, RNA samples were analyzed on an Agilent 2100 Bioanalyzer (Agilent) with the Agilent RNA 6000 Nano Kit (Agilent) and DS-11 FX+ Spectrophotometer (DeNovix) and stored at– 70˚C.

Library construction and sequencing of samples were performed by the CEITEC Genomics core facility (Brno, Czechia) on Illumina NextSeq 500, single-end, 75 bp.

## Bioinformatics analysis

RNA-Seq data consisted of four replicates (B, C, D, and E), as described in our previous papers [20,21]. The data are available from the NCBI Sequence Read Archive (SRA) under the accession number SRP033480. During preprocessing of data adapters, trimming was conducted by Trimmomatic [22]. Reads corresponding to 16S and 23S rRNA were removed with SortMeRNA [23] together with the SILVA database [24] containing sequences of known bacterial 16S and 23S genes. The *C. beijerinckii* NRRL B-598 (CP011966.3) genome was used as the reference for mapping of preprocessed reads and mapping was performed by STAR [25].

Count tables of mapped reads were estimated in R/Bioconductor by featureCounts function from the Rsubread package [26]. Differential expression analysis was conducted by the DESeq2 package[27] in R. This package contains functions for normalization of raw count tables by library size. Normalized count tables were used for evaluation of transcription profiles of selected genes by Z-scores that were visualized as heatmaps by the R packages gplots

and RColorBrewer. RPKM values were estimated by the featureCounts function from the edgeR package [28] in R/Bionconductor. Time series plots were generated in Matlab R2017a.

### Identifications of genes and homology search

Genes encoding the uptake of amino acids, sugars, vitamins and metal ions were identified based on key word searches in these databases: UniProt [29], InterPro [30] and PFAM [31]. Most of the genes were verified by BLASTP [32] searches against the non-redundant (nr) database with experimentally verified or automatically predicted proteins encoding transport in *C. beijerinckii* NCIMB 8052, *C. acetobutylicum* ATCC 824, other solventogenic and pathogenic clostridial species, *Bacillus* species or other model bacteria.

## Results and discussion

The experiments comprised two-phase ABE fermentation: the acidogenic phase, which is characterized by acid production and pH decrease, and the solventogenic phase, when glucose and organic acids are utilized to produce solvents. The shift from acidogenesis (sampling point T1) to solventogenesis (sampling points T3-T6) occurred at approximately 6 h of cultivation (sampling point T2) (Fig 1B). Granulose accumulation was observed at 8.5 h of cultivation (sampling point T3), spore formation began at approximately 13 h of cultivation (sampling point T4) and the first mature spores were observed at 23 h (sample point T6) (Fig 1C). Sampling point T5 represented the beginning of the stationary phase (Fig 1A). More detailed information about cell physiology and production rates during the fermentation can be found in Sedlar *et al.* (2018) and Patakova *et al.* (2019) [20,21].

### Amino acid uptake

About 80 genes encoding putative amino acid transporters were identified in the genome of *C. beijerinckii* NRRL B-598, based on key word searches in protein function databases and verified by BLASTP searches against protein sequences encoding transporters described in other species (S1 File, S2 File). Only seven genes encoding amino acid uptake were not differentially expressed throughout the course of cultivation.

Tryptone and yeast extract, as a part of TYA medium, represented the universal, non-specific source of amino acids in this experiment. Solventogenic clostridia are sometimes considered to be amino acid auxotrophs [11] and some of them are not able to grow without addition of yeast extract to the medium [33]. However, different *C. acetobutylicum* strains, including *C. acetobutylicum* P262 later reclassified to *Clostridium saccharobutylicum NCP262*, were successfully cultured in defined, synthetic medium [34–36]. According to the latest *in silico* analysis, genes encoding all essential enzymes for proteinogenic amino acid synthesis are present in the *C. beijerinckii* G117 genome [37]. However, uptake of free amino acids from the culture medium is energetically more advantageous for the cell than their biosynthesis [38].

For most differentially expressed genes encoding amino acid uptake, the highest level of expression was set at the beginning of cultivation (T1) (S2 File), which is probably connected to the use of amino acids for growth. Moreover, amino acids take part in the stress response to acids produced in this phase [39], during which, *C. acetobutylicum* cells have been shown to accumulate the highest intracellular concentration of amino acids [40].

Branched-chain amino acids BCAA (leucine, isoleucine and valine) are essential for adaptation to solvent formation [41]. They are precursors of membrane fatty acids and therefore take part in stress response mechanisms [42]. For *Clostridioides difficile*, BCAA are described as being essential and growth-limiting [43,44].The most massive increase in the expression of genes encoding uptake of BCAA, X276_18220- X276_18200 and X276_00370—X276_00350,

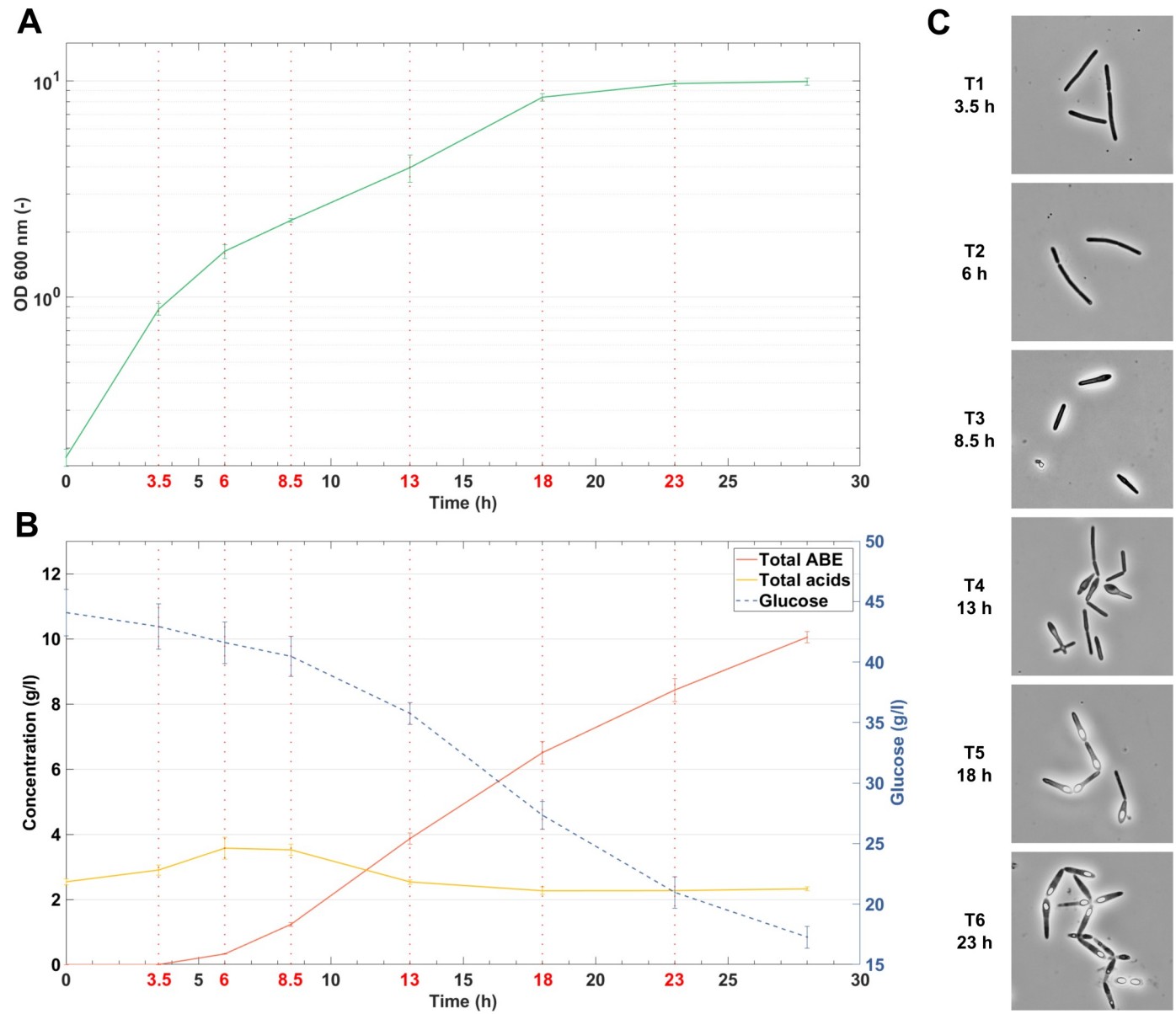

**Fig 1. Fermentation profile of *C. beijerinckii* NRRL B-598 during ABE fermentation on TYA medium.** (A) Growth curve. (B) The concentration of glucose, solvents and acids. (C) Cell morphology at the moment of sampling for RNA-Seq (magnification 1000x). Values are the means and standard deviations of the two biological replicates. Sampling points for RNA-Seq analysis are indicated with red vertical dotted lines and/or by red text labels.

was observed between T2 and T3, corresponding to the beginning of solventogenesis (Fig 2, S1 File). According to previous findings, genes responsible for fatty acid biosynthesis in *C. beijerinckii* NRRL B-598 showed the highest expression profile between T3 to T4 following BCAA uptake [20].

Glutamine transport genes X276_14095, X276_05000-X276_04985 were also differentially expressed during solventogenesis (S1 File). This amino acid has been reported to help *C. beijerinckii* SA-1 survive butanol stress [45]. Moreover, for *C. beijerinckii* NCP 260, its addition is reported to have an impact on butanol titer. As glutamine is converted to glutamate, it

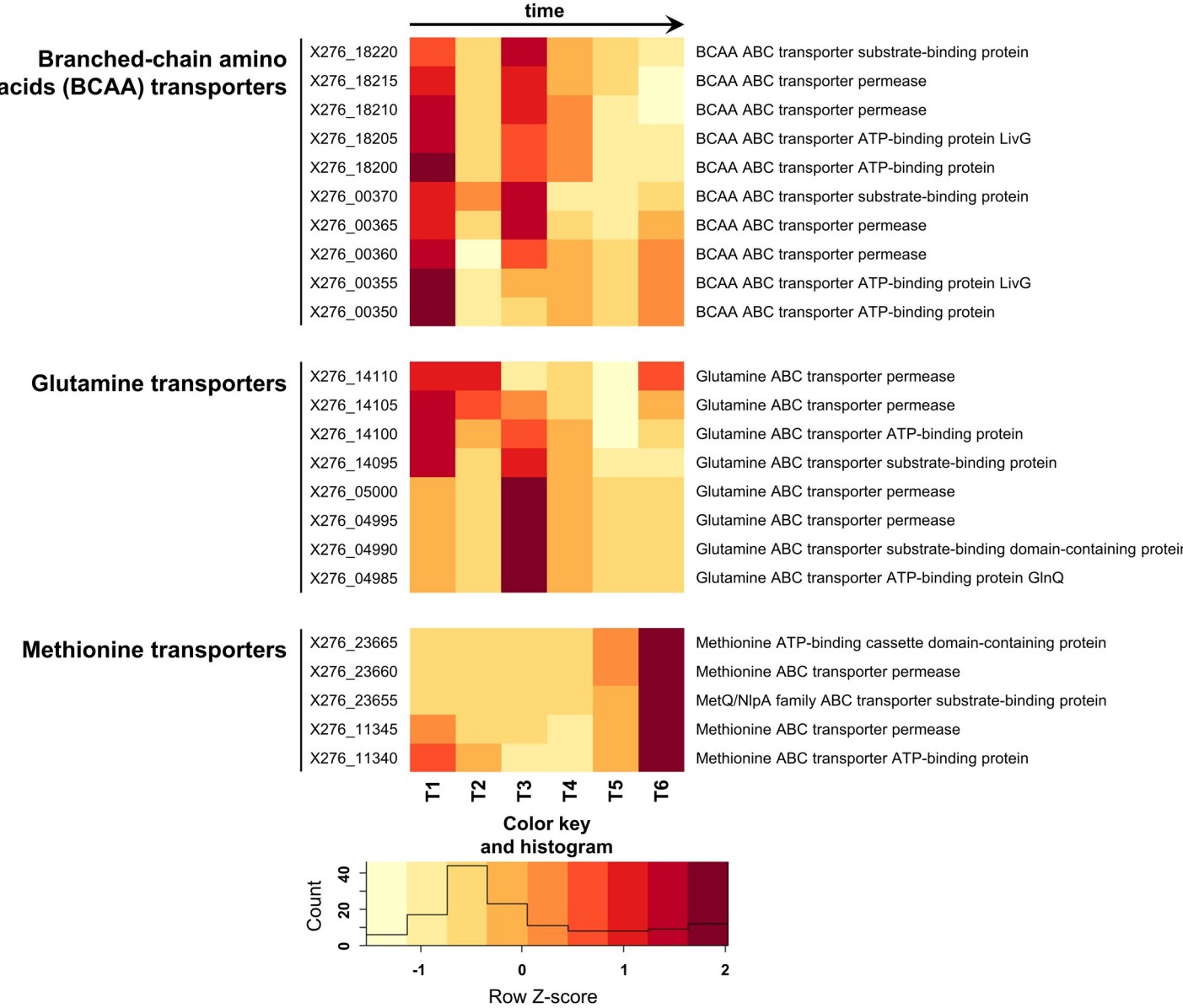

**Fig 2. Heatmap displaying changes in transcriptions of the selected genes encoding putative amino acid transporters.**

indirectly enhances the acid tolerance response of the cell, which leads to an increase in viable cells that enter solventogenesis and, as a consequence, higher butanol titer [15].

Genes encoding methionine uptake systems, X276_23665- X276_23655 and X276_11345-X276_11340, were differentially expressed (padj < 0.001, Benjamini-Hochberg correction) between late adjacent sampling points, i.e. between T4 and T5 and between T5 and T6 (S1 File). There is evidence that intracellular accumulation and secretion, as well as extracellular addition of methionine can aid survival under stress conditions and leads to a slightly improved rate of butanol synthesis in *C. acetobutylicum* [7]. Moreover, methionine is probably responsible for upregulation of butanol synthesis in the strain [46]. Previously described as a putative non-PTS glucose transporter gene, X276_11345, [20] after in-depth analysis, was re-identified as methionine ABC transporter permease.

The greatest number of amino acid transporters belong to the ATP-binding cassette transporter family (ABC transporter). However, amino acid uptake can also occur via, for example, sodium:solute symporters X276_22340 and X276_19775 or sodium:dicarboxylate symporter X276_03860, which is discussed below.

## Metal ion and vitamin transport

We identified 81 genes encoding metal ion and vitamin transporters in the *C. beijerinckii* NRRL B-598 genome. Thirteen of them were not differentially expressed during cultivation (S4 File).

Tryptone and yeast extract were the source of vitamins and trace metals in the medium, and potassium, sodium, magnesium and iron were added in the form of potassium dihydrogenphosphate, sodium hydroxide (pH adjustment), magnesium sulfate heptahydrate and ferrous sulfate heptahydrate.

Potassium and sodium, among other functions, play crucial roles in cell homeostasis and membrane transport processes. Accumulation of potassium in the cell has an effect on membrane stability, cell integrity and division. In *C. beijerinckii* NRRL B-598, potassium is probably transported via the Kdp system, X276_26900-X276_19380-X276_19375-X276_19370, or via one of the potassium transport system Kup family proteins (Fig 3, S3 File). Transport of sodium usually happens via a symport or antiport, and, in the case of symport, can be associated with the uptake of amino acids, sugars, organic cations or anions. The sodium:dicarboxylate symporter X276_03860 shares a high similarity with the sodium:glutamate/aspartate symporter CD630_25410 of *C. difficile* 630, with a high specificity for aspartate [47]. Potassium and sodium ions are also important components of the germination process [48], which may explain the high level of expression of some of their transporters during stationary phase, when spores are formed. A high intracellular potassium concentration was found to be important for sporulation in *Bacillus subtilis* [49].

The strongest expression of magnesium transporters, X276_26115, X276_17370 and X276_09335, was during the transition period between acidogenesis and solventogenesis (T2) (Fig 3, S3 File). Except for its function in cellular energetics i.e. to form a chelate with adenosine triphosphate (ATP) and to take part in ATP-dependent reactions [50], magnesium is essential for the functioning of acetate kinase [51]. Acetate kinase gene X276_20705, encoding the enzyme converting acetyl-CoA to acetate, was highly expressed at the beginning of cultivation and was differentially expressed between T1 and T2, reflecting its role in acidogenesis (S7 File). It is also reported that magnesium ions take part in stabilization of the cell membrane as a stress response to solvent production for *C. beijerinckii* RZF-1108 [52].

The ferrous uptake system *feo*, encoded by X276_27330-X276_04855-X276_04850-X276_04845, is responsible for anaerobic iron uptake [41] and was highly expressed during the course of cultivation, with a local maxima at time T1 (Fig 3, S3 File). Iron is a component of ferredoxin, the iron-sulfur protein involved in electron transport. It is also a part of [FeFe]-hydrogenase and [NiFe]-hydrogenase, which take part in hydrogen formation via catalysis of proton reduction. For *C. acetobutylicum*, the main production of hydrogen occurs during acidogenesis [53] and the highest expression of iron transporters can be connected with a high iron requirement for activity of ferredoxin and hydrogenases, which play important roles in the pathway [50,52,54]. It is reported that iron addition facilitates hydrogen biosynthesis in *C. beijerinckii* IB4 [55]. At the same time, similarly to *C. acetobutylicum* [41], other ferrous uptake systems, X276_13995-X276_13990-X276_13985 and X276_11140-X276_11135, showed increased expression during stationary phase (Fig 3, S3 File), which may be due to the role of Fe in sporulation. It is reported that for *Bacillus* spp., iron is a sporulation inducer [56].

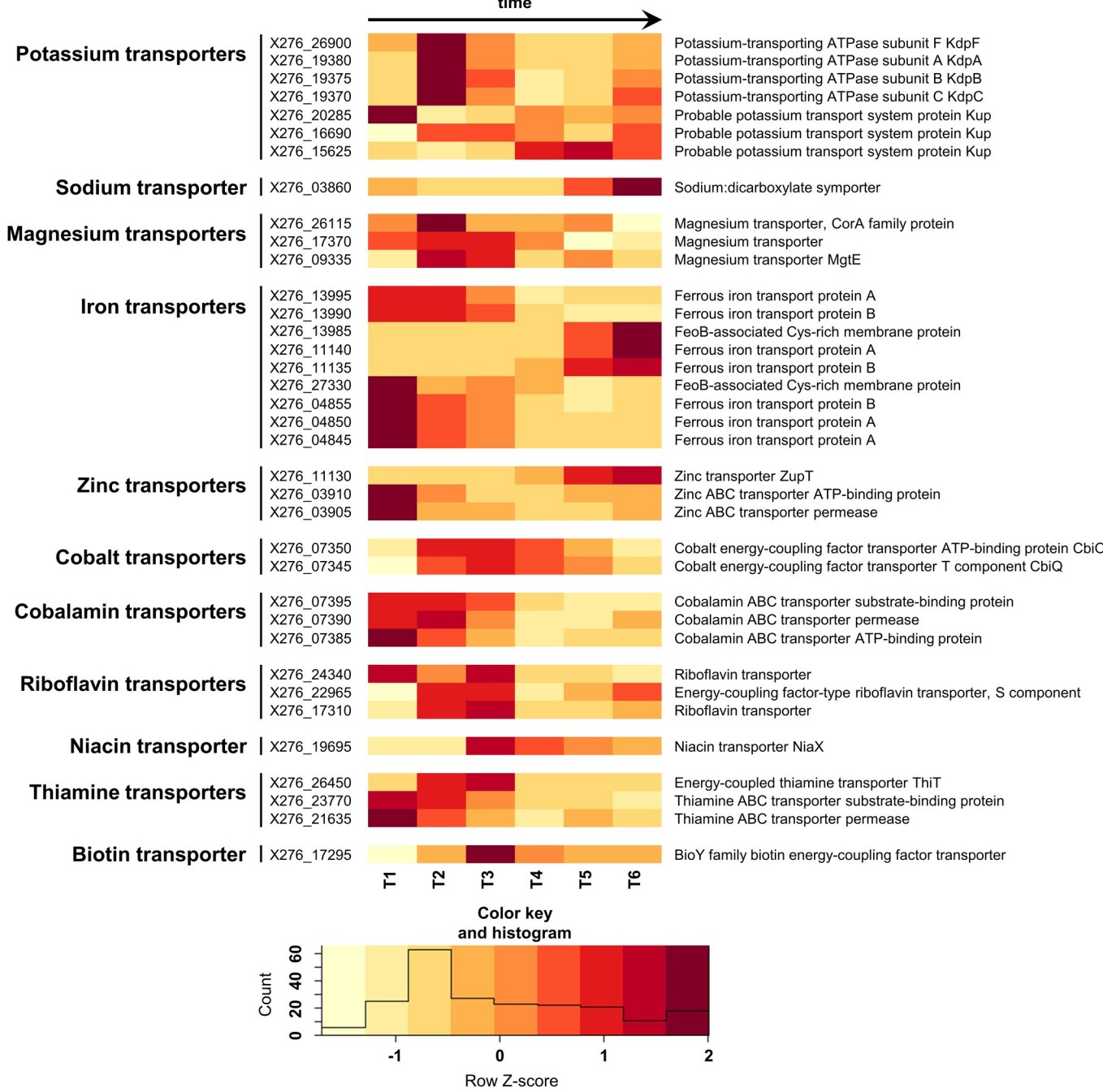

**Fig 3. Heatmap displaying changes in transcriptions of the selected genes encoding putative metal ion and vitamin transporters.**

Zinc ions are involved in the metabolism of proteins, nucleic acids, carbohydrates and lipids. Zinc is a cofactor of metalloenzymes such as alkaline phosphatase, alcohol dehydrogenase and aminopeptidase. Yeast extract was the source of zinc ions in this experiment. The highest level of expression of zinc ABC transporter X276_03910-X276_03905 was established at the beginning of cultivation (T1) (Fig 3, S3 File). This seems to be connected with the function of zinc in stimulating cell growth and regulating sugar transport [8,50]. On the other hand, zinc

transporter ZupT, encoded by X276_11130, was differentially expressed during solventogenesis (S3 File), which may be linked with a higher zinc requirement for the functioning of butanol dehydrogenase, a key enzyme responsible for butanol biosynthesis in clostridia. Additional supplementation of zinc to the medium is reported to increase solvent formation and initiate earlier solventogenesis in *C. acetobutylicum*, probably due to enhancement of butanol dehydrogenase activity [8]. Moreover, zinc supplementation facilitates acid and butanol tolerance in *C. acetobutylicum* [57].

Cobalt participates in enzyme-catalyzed hydrolytic and redox reactions. It is also a part of the structure of vitamin $B_{12}$ cobalamin, which is involved in methyl transfer reactions and fatty acid catabolism. Most microorganisms are not able to synthesize cobalamin [58], but it is reported that the *C. acetobutylicum* genome contains genes for its biosynthesis [59]. Generally, metal ions (including cobalt) are transported from an environmentally bioavailable source and are an essential part of the culture medium. Bacteria are able to synthesize vitamins, nevertheless, vitamin uptake is energetically more advantageous for the cell and is also very important when an organism lacks a full biosynthetic pathway [60,61].

Genes encoding putative transporters of cobalt and cobalamin were differentially expressed at T1-T2 and T3-T4 respectively (S3 File). Addition of cobalt positively affected sugar conversion efficiency and production of hydrogen during acidogenesis in *C. acetobutylicum* NCIM 2877 [62]. Cobalamin, on the other hand, may ameliorate the effect of toxic compounds in the medium, such as furfural [63] or probably produced solvents.

Riboflavin $B_2$ transporters X276_24340, X276_22965 and X276_17310 had significant changes in their expression (padj < 0.001, Benjamini-Hochberg correction) between T1 and T2 and between T3 and T4 (S3 File). Riboflavin is a precursor of flavin mononucleotide and flavin adenine dinucleotide, formation of which is catalyzed by riboflavin kinase. Riboflavin kinase *rfk* X276_20490 exhibited increased transcription during T1 and was highly expressed during solventogenesis with a local maximum at time T4 (S7 File). It seems that transported riboflavin was sequentially used in flavin formation. Because flavins can act as electron carriers and cofactors for the redox reactions, it can be concluded that changes in expression of riboflavin transporters were connected with maintenance of the cellular redox balance. This function of the vitamin was also hypothesized in *C. beijerinckii* NCIMB 8052 under furfural stress [63]. Additionally, it was shown that riboflavin play role in iron acquisition in *Helicobacter pylori*, *Campylobacter jejuni*, *Shewanella* and probably *C. acetobutylicum* [64–67], however, expression of riboflavin transporters did not correspond with the maxima of expression of iron transporters (see above).

Niacin $B_3$ transporter *niaX* X276_19695 was differentially expressed between all adjacent sampling points expect for insignificant change between T5 and T6 (S3 File). Homologues of the gene encoding the NiaX transporter, can be found in multiple *Streptococci* and *Clostridium* [68]. Niacin is the precursor of NADH and NADPH, and its addition to the medium is reported to increase butanol yield and productivity in *Clostridium* sp. strain BOH3 due to flux redistribution [69].

Thiamin $B_1$, biotin $B_7$ and para-aminobenzoic acid $B_{10}$ (PABA) are specific vitamins of some of the defined media used for the cultivation of solventogenic clostridia [34–36]. Genes encoding the uptake of thiamin and biotin were highly expressed between T1 and T4, reflecting their role in sugar uptake and metabolism (Fig 3, S3 File). B class vitamins are described as limiting factors for the performance of solventogenic clostridia and for efficient ABE fermentation. Overexpression of genes encoding transport and biosynthesis of thiamine and biotin in *C. acetobutylicum* improved growth and sugar utilization rates and increased the solvent titer [9,10]. PABA takes part in folate $B_9$ biosynthesis, although the transport of PABA has still not been described in bacteria. However, genes similar to ones encoding its biosynthesis in

*Lactococcus lactis* have been identified in the genome of *C. beijerinckii* NRRL B-598: glutamine amidotransferase *pabA* X276_22125 and X276_10695, aminodeoxychorismate synthase *pabB* X276_05255 and aminodeoxychorismate lyase *pabC* X276_05260 (S7 File).

## Carbohydrate uptake via PTS and non-PTS transporters

In *C. beijerinckii* NRRL B-598, glucose, the main carbohydrate used for the described experiments, is transported into the cell via the PTS. Transport by PTS involves enzymes that are situated in both the cell membrane—integral membrane sugar permease (IIC/IID)—and the cytoplasm: phosphoryl transfer protein enzyme I (PtsI), histidine-containing protein (PtsH), enzyme IIA (EIIA) and enzyme IIB (EIIB) (Fig 4). The genome of *C. beijerinckii* NRRL B-598 includes one copy of each gene encoding the histidine-containing protein PtsH (X276_20425) and enzyme I protein PtsI (X276_25680). It is common in clostridia that genes encoding PtsH and PtsI are located in different parts of the genome, while enzyme II subunits are clustered together and are organized into operons [70].

About 40 sets of the PTS EII genes were identified in the *C. beijerinckii* NRRL B-598 genome, with the highest number of genes encoding cellobiose and mannose uptake (S5 File, S6 File). This amount is comparable with *C. beijerinckii* NCIMB 8052 and is much higher than *C. acetobutylicum* ATCC 824, with 47 and 14 sets respectively [71]. This reflects the ability of *C. beijerinckii* strains to utilize a wide range of substrates as carbon sources and demonstrates their metabolic flexibility [70]. We determined experimentally that *C. beijerinckii* NRRL B-598 is able to utilize glucose, xylose, arabinose, mannose, saccharose, cellobiose, galactose and pectin and is not able to grow on lactose and glycerol.

Eight genes encoding the glucose family PTS EII proteins–orthologues of glucose PTS EII genes of *C. acetobutylicum* ATCC 824 [13,72] and *C. beijerinckii* NCIMB 8052 [14,71,73]–were identified in the *C. beijerinckii* NRRL B-598 genome.

Most of the glucose family PTS EII genes demonstrated low amount of mapped reads, some of them even neglectable/at the noise threshold (S5 File). Therefore, they were probably of minor significance for glucose uptake. The genes encoding *gluIIA-gluIIBC* (X276_03050-X276_03055) presumably were the exception. These genes exhibited increase in their transcription, including statistically significant differential expression (padj < 0.001, Benjamini-Hochberg correction) between T2 and T3 and seems to be involved in glucose transport at the moment (Fig 5, S5 File).

Most of the other sugar transporters (PTS EII subunits and non-PTS systems) were not expressed during T1-T3, except for mannose PTS EII genes X276_23245- X276_23235 and X276_02990- X276_02975 (Fig 5, S5 File, S7 File). Mannose EII-encoding genes were the ones with the highest number of mapped reads in the whole transcriptome and were massively expressed during both acidogenesis and solventogenesis, taking part in glucose uptake [20]. Such massive involvement of the mannose PTS EII genes in glucose transport, as a contrary to glucose PTS EII genes, was also observed in other *C. beijerinckii* [74–76]. According to previous research, mannose PTS EII genes are responsible for uptake and phosphorylation of multiple carbohydrates, in particular, mannose, glucose, fructose and sorbose [75].

About one third of the carbohydrate transport genes were not differentially expressed during ABE fermentation, because there was no substrate for them to uptake. On the other hand, fructose, galactose and mannitol PTS EII (Fig 5, S5 File) and galactose ABC transporter genes (S6 File), which also did not have substrate to uptake, had a relatively high number of mapped reads and were differentially expressed during cultivation. Expression of fructose transporter X276_17630 was decreasing continuously for several hours when all pairwise comparison of adjacent sampling times within the interval T1–T4 were evaluated as significant differential

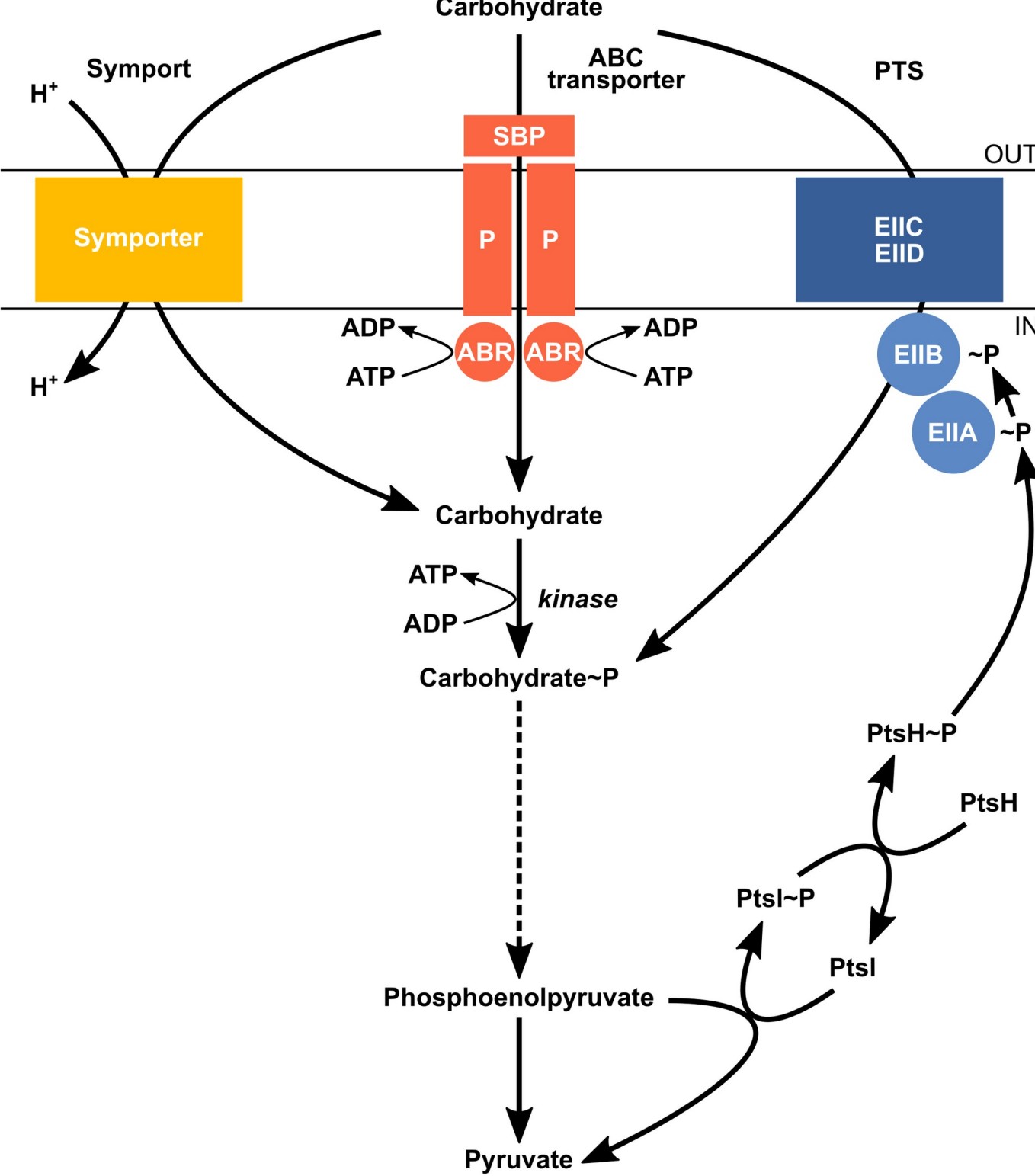

**Fig 4. A simplified scheme of carbohydrate uptake in *C. beijerinckii* NRRL B-598.** Carbohydrate uptake can be carried out via three types of membrane transporters: electrochemical potential-driven transporters (symport), primary active transporters (ATP-binding cassette transporters ABC: SBP–substrate-binding protein; P–carrier protein; ABR–ATP-binding region) and group translocators (Phosphoenolpyruvate -dependent phosphotransferase system PTS: EIIC/EIID—integral membrane sugar permeases; EIIB–enzyme IIB; EIIA—enzyme IIA; PtsH—histidine-containing protein; PtsI—phosphoryl transfer protein enzyme I).

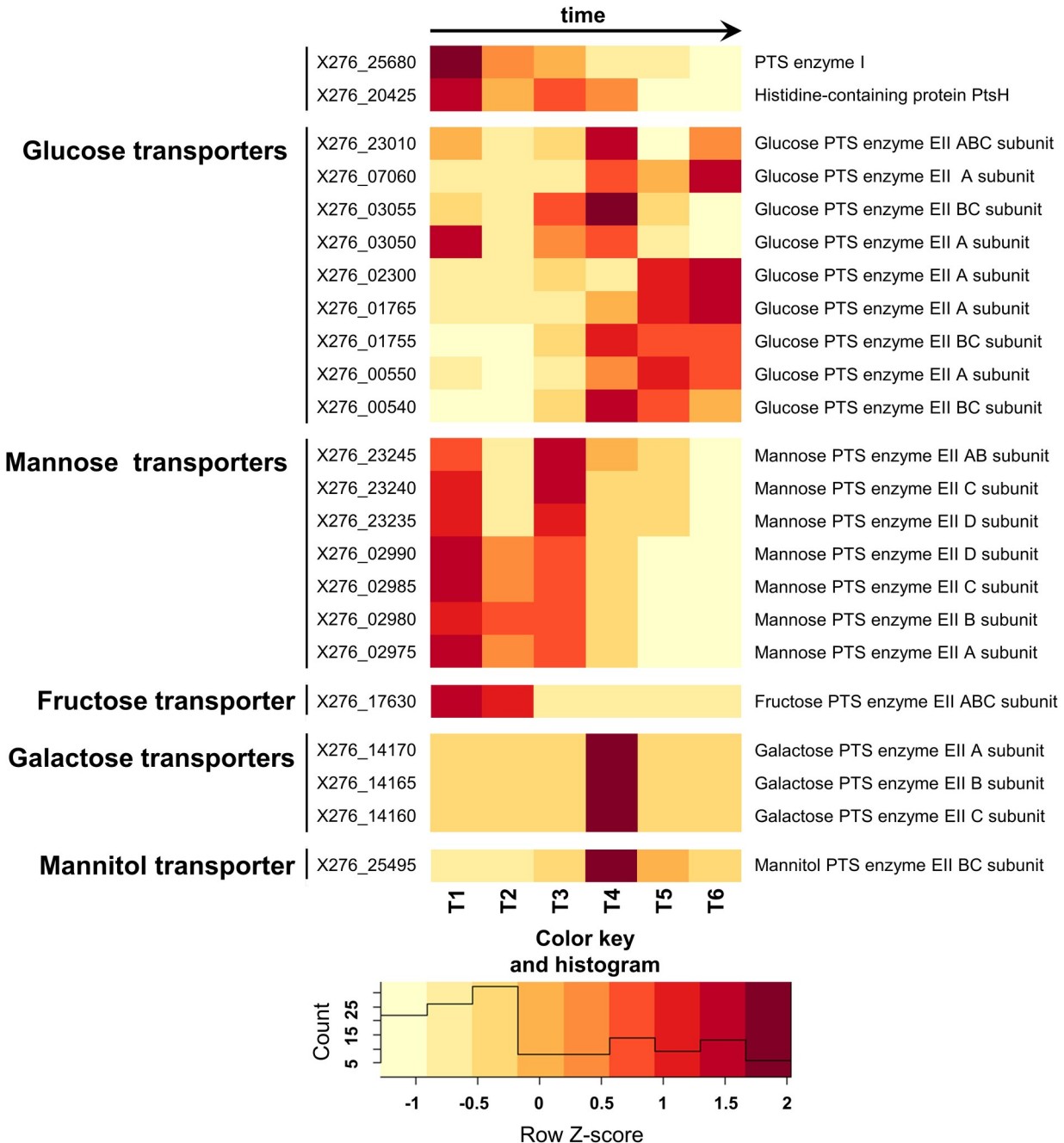

**Fig 5. Heatmap displaying changes in transcriptions of the selected genes encoding putative carbohydrate transporters.**

expression (padj < 0.001, Benjamini-Hochberg correction). It is reported for *C. acetobutylicum* ATCC 824, that expression of the fructose uptake genes can be induced by the presence of glucose in the medium [13]. As for galactose or mannitol transporters, differential expression probably was not induced by the presence of substrate. We determined the expression of genes encoding enzymes that are involved in the transport these carbohydrates in respective pathways, for example, galactokinase X276_03575, galactose-1-phosphate uridylyltransferase

X276_03565 or mannitol-1-phosphate 5-dehydrogenase X276_25480, but they were either not differentially expressed during cultivation or their differential expression did not logically follow expression of transport genes (S5 File, S7 File). Differential expression of these transport genes may be caused by general metabolic changes connected with sporulation or butanol stress, nevertheless, confirmation requires further investigation.

Although, carbohydrates are mostly transferred to clostridial cells through PTS, sugar uptake partially occurs by non-PTS systems, which are represented by ABC transporters and symporters (Fig 4) [12,77]. Putative xylose symporter genes *xylT* X276_26120 and *xynB* X276_05225, similar to those found in *C. acetobutylicum* ATCC 824 [13], as well as xylose ABC transporter XylFGH, similar to the ABC-type D-xylose transporter found in *C. beijerinckii* NCIMB 8052 [78], coded by X276_14760- X276_14750, were identified in the *C. beijerinckii* NRRL B-598 genome. None of these non-PTS systems were differentially expressed during fermentation (S6 File), however, these genes may be of major importance when the strain is cultivated on agricultural waste.

Hexoses that are taken up enter glycolysis, pentoses enter the pentose phosphate pathway and then glycolysis, other sugars are converted into intermediate products of the pathways and are further transformed into pyruvate [79]. More information about the expression of genes encoding enzymes that take part in the central metabolism of *C. beijerinckii* NRRL B-598 can be found in Patakova *et al.* (2019) [20].

## Conclusions

The data presented here combine the identification and analysis of transcriptomic profiles of genes encoding putative amino acid, metal ion, vitamin and carbohydrate transporters covering the whole life cycle of *C. beijerinckii* NRRL B-598. To our best knowledge, this article is the first to comprehensively describe genes encoding the uptake of the main nutrients in butanol-producing *C. beijerinckii* strain.

Our results suggest that transcriptomic data of genes encoding nutrient uptake may be used to predict an increased requirement for these substances during different phases of ABE fermentation. From the transcriptomic data obtained for *C. beijerinckii* NRRL B-598 in TYA medium, we can assume that during acidogenic phase strain exhibits increased requirement for such nutrients as branched-chain amino acids (leucine, isoleucine and valine), glutamine, iron, zinc, cobalamin, riboflavin and thiamin. During the shift from acidogenesis to solventogenesis/ beginning of the solventogenic phase strain seemed to require branched-chain amino acids (leucine, isoleucine and valine), glutamine, magnesium, cobalt, cobalamin, riboflavin, thiamin and biotin. During solventogenic phase such nutrients as methionine, potassium, sodium, iron, zinc and niacin were probably very important for the functioning of the strain.

We recommend addition of glutamine, methionine, zinc, niacin, thiamine and biotin to the culture medium for the improvement of ABE fermentation performance, especially, when the strain is cultivated on waste substrates that do not contain these growth factors. These nutrients were of high demand for the strain, but more importantly, according to the literature, they had a positive impact on butanol titer.

Bacteria are able to synthesize most of the nutrients; however, uptake from the culture medium is energetically more advantageous for the cell. It is possible that energy surplus can be used for the production of valuable metabolites, synthesis of which are less energetically advantageous for the cell, for example, solvents in case of in *C. beijerinckii* NRRL B-598.

We hope that the present study stimulates further investigations of transport systems in solventogenic clostridia, which will lead to efficient optimization of the culture medium and selection of the best production strains for biobutanol production.

## Supporting information

**S1 File. Additional data on genes encoding putative amino acid transporters from Fig 2.** A. RPKM values. B. Differential expression analysis.
(XLSX)

**S2 File. Other genes encoding putative amino acid transporters in *C. beijerinckii* NRRL B-598.** A. RPKM values. B. Differential expression analysis. C. Heatmap.
(XLSX)

**S3 File. Additional data on genes encoding putative metal ion and vitamin transporters from Fig 3.** A. RPKM values. B. Differential expression analysis.
(XLSX)

**S4 File. Other genes encoding putative metal ion and vitamin transporters in *C. beijerinckii* NRRL B-598.** A. RPKM values. B. Differential expression analysis. C. Heatmap.
(XLSX)

**S5 File. Additional data on genes encoding putative carbohydrate transporters from Fig 5.** A. RPKM values. B. Differential expression analysis.
(XLSX)

**S6 File. Other genes encoding putative carbohydrate transporters in *C. beijerinckii* NRRL B-598.** A. RPKM values. B. Differential expression analysis. C. Heatmap.
(XLSX)

**S7 File. Transcriptomic data for other genes mentioned in this article.** A. RPKM values. B. Differential expression analysis. C. Heatmap.
(XLSX)

## Acknowledgments

We acknowledge the CESNET LM2015042 and the CERIT Scientific Cloud LM2015085 for providing computational resources under the program "Projects of Large Research, Development, and Innovations Infrastructures". We acknowledge the CF Genomics of CEITEC, which is supported by the NCMG research infrastructure (LM2015091 funded by MEYS CR), for their support in obtaining the scientific data presented in this paper.

## Author Contributions

**Conceptualization:** Maryna Vasylkivska, Katerina Jureckova, Barbora Branska, Jan Kolek, Ivo Provaznik, Petra Patakova.

**Data curation:** Karel Sedlar.

**Formal analysis:** Katerina Jureckova.

**Funding acquisition:** Ivo Provaznik, Petra Patakova.

**Investigation:** Maryna Vasylkivska, Barbora Branska, Jan Kolek.

**Project administration:** Ivo Provaznik, Petra Patakova.

**Software:** Karel Sedlar.

**Supervision:** Petra Patakova.

**Validation:** Maryna Vasylkivska, Barbora Branska, Karel Sedlar, Jan Kolek.

**Visualization:** Katerina Jureckova.

**Writing – original draft:** Maryna Vasylkivska.

**Writing – review & editing:** Katerina Jureckova, Barbora Branska, Karel Sedlar, Jan Kolek, Ivo Provaznik, Petra Patakova.

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
