## [Decision Letter · Decision Letter 0]

11 Sep 2019

PONE-D-19-22508

Transcriptional analysis of amino acid, metal ion, vitamin and carbohydrate uptake in butanol-producing Clostridium beijerinckii NRRL B-598

PLOS ONE

Dear Mrs. Vasylkivska,

Thank you for submitting your manuscript to PLOS ONE. After careful consideration, we feel that it has merit but does not fully meet PLOS ONE’s publication criteria as it currently stands. Therefore, we invite you to submit a revised version of the manuscript that addresses the points raised during the review process.

We would appreciate receiving your revised manuscript by Oct 26 2019 11:59PM. To enhance the reproducibility of your results, we recommend that if applicable you deposit your laboratory protocols in protocols.io, where a protocol can be assigned its own identifier (DOI) such that it can be cited independently in the future. For instructions see: http://journals.plos.org/plosone/s/submission-guidelines#loc-laboratory-protocols

We look forward to receiving your revised manuscript.

Kind regards,

Zhiqiang Wen, Ph.D.

Academic Editor

PLOS ONE

Journal Requirements:

Reviewers' comments:

Reviewer's Responses to Questions

**Comments to the Author**

1. Is the manuscript technically sound, and do the data support the conclusions?

Reviewer #1: Yes

Reviewer #2: Yes

2. Has the statistical analysis been performed appropriately and rigorously? 

Reviewer #1: Yes

Reviewer #2: Yes

3. Have the authors made all data underlying the findings in their manuscript fully available?

Reviewer #1: Yes

Reviewer #2: Yes

4. Is the manuscript presented in an intelligible fashion and written in standard English?

Reviewer #1: Yes

Reviewer #2: Yes

5. Review Comments to the Author

Reviewer #1: Clostridium beijerinckii are important industrial microorganisms as they possess the ability to produce solvents including acetone-butanol-ethanol during the fermentation (also called ABE fermentation). Illumination the metabolism of C. beijerinckii can accelerate the research of ABE fermentation. In this manuscript, authors collected the transcriptomic data of the genes obtained during the whole life of C. beijerinckii NRRL B-598. Based on these data, authors analyzed the uptake of main nutrients including amino acid, metal ion, vitamin and carbohydrate in C. beijerinckii NRRL B-598. This research may provide guidance for optimization of the culture medium to improve the biobutanol production.

Other comments:

1. Please conclude the detailed guidance for improving the ABE fermentation performance in the conclusion part.

2. Can the transcriptional analysis of C. beijerinckii NRRL B-598 in YTA medium represent the situations in other medium?

Reviewer #2: This manuscript comprehensively described genes encoding the uptake of the main nutrients in butanol-producing C. beijerinckii strain. I suggest its publication after addressing minor points:

1. Line 120 to 124, it need appropriate additions to HPLC conditions such as temperature and flow rate.

2. line 336, ‘may be involved in the maintenance of the cellular redox balance during acidogenesis and the beginning of solventogenesis.’ What led to this result and how to understand it?

3. line 394, ‘All of these genes were probably of minor significance for glucose uptake and were slightly expressed only during later stages of solventogenesis.’ The detail should be explained.

6. PLOS authors have the option to publish the peer review history of their article (what does this mean?). If published, this will include your full peer review and any attached files.

Reviewer #1: No

Reviewer #2: Yes: Dali Li

---

## [Author Response · Author response to Decision Letter 0]

30 Sep 2019

Response to reviewers’ comments:

Reviewer #1

Clostridium beijerinckii are important industrial microorganisms as they possess the ability to produce solvents including acetone-butanol-ethanol during the fermentation (also called ABE fermentation). Illumination the metabolism of C. beijerinckii can accelerate the research of ABE fermentation. In this manuscript, authors collected the transcriptomic data of the genes obtained during the whole life of C. beijerinckii NRRL B-598. Based on these data, authors analyzed the uptake of main nutrients including amino acid, metal ion, vitamin and carbohydrate in C. beijerinckii NRRL B-598. This research may provide guidance for optimization of the culture medium to improve the biobutanol production.

We are grateful for your comments.

Other comments:

1. Please conclude the detailed guidance for improving the ABE fermentation performance in the conclusion part.

We extended the conclusion part of the manuscript and added information about individual nutrients that were required by C. beijerinckii NRRL B-598 during each phase of ABE fermentation: acidogenic phase, shift from acidogenesis to solventogenesis/ beginning of the solventogenesis and solventogenic phase. Nutrients that were of an increased demand for the strain and which were described in literature as ones with the positive impact on butanol titer were also highlighted.

2. Can the transcriptional analysis of C. beijerinckii NRRL B-598 in YTA medium represent the situations in other medium?

It seems that increased demand for branched amino acids, selected trace metals and other nutrients during particular phases of ABE fermentation might be a general feature. However, further experiments are necessary for the confirmation.

Because we were also interested in this question, some of our recent experiments included transcriptional analysis of C. beijerinckii NRRL B-598 in different medium (RCM medium). RCM medium differs in composition with the TYA medium used for the experiments described in the article. Interestingly, when cultivated in the RCM medium C. beijerinckii NRRL B-598 exhibits non-sporulating phenotype [1] and is more prone to acid crash phenomena. We are planning separate publication for the results of this transcriptional analysis and part of the publication will include comparison of the results obtained in both of media.

Reviewer #2

This manuscript comprehensively described genes encoding the uptake of the main nutrients in butanol-producing C. beijerinckii strain. I suggest its publication after addressing minor points:

We would like to thank you for your suggestions.

1. Line 120 to 124, it need appropriate additions to HPLC conditions such as temperature and flow rate.

More detailed information about HPLC conditions was added to the manuscript.

2. line 336, ‘may be involved in the maintenance of the cellular redox balance during acidogenesis and the beginning of solventogenesis.’ What led to this result and how to understand it?

We agree that more explanation was needed regarding this result. 

Riboflavin is a precursor of flavin mononucleotide and flavin adenine dinucleotide, formation of which is catalyzed by riboflavin kinase. Riboflavin kinase rfk X276_20490 exhibited increased transcription during T1 and was highly expressed during solventogenesis with a local maximum at time T4. It seems that transported riboflavin was sequentially used in flavin formation. Because flavins can act as electron carriers and cofactors for the redox reactions, it can be concluded that changes in expression of riboflavin transporters were connected with maintenance of the cellular redox balance. This function of the vitamin was also hypothesized in C. beijerinckii NCIMB 8052 under furfural stress [2].

We added this explanation to the article and the other potential function of riboflavin was discussed in the paragraph. Transcriptomic data on riboflavin kinase rfk X276_20490 was added to S7 File.

3. line 394, ‘All of these genes were probably of minor significance for glucose uptake and were slightly expressed only during later stages of solventogenesis.’ The detail should be explained.

We stated that glucose family PTS EII genes were probably of minor significance for glucose uptake because most of the genes demonstrated low amount of mapped reads, some of them even neglectable/at the noise threshold (S5 File). We also presumed that glucose PTS EII genes were not involved in glucose uptake because from the transcriptomic data it was obvious that mannose PTS EII genes were much more involved in the transport. Involvement of the mannose PTS EII genes in glucose transport, as a contrary to glucose PTS EII genes, was also observed in other C. beijerinckii [3–5]. Related paragraph was changed and more details were added.

1. Kolek J, Branska B, Drahokoupil M, Patakova P, Melzoch K. Evaluation of viability, metabolic activity and spore quantity in clostridial cultures during ABE fermentation. FEMS Microbiol Lett. 2016;363: fnw031. doi:10.1093/femsle/fnw031

2. Zhang Y, Ezeji TC. Transcriptional analysis of Clostridium beijerinckii NCIMB 8052 to elucidate role of furfural stress during acetone butanol ethanol fermentation. Biotechnol Biofuels. 2013;6: 66. doi:10.1186/1754-6834-6-66

3. Wang Y, Li X, Mao Y, Blaschek HP. Genome-wide dynamic transcriptional profiling in Clostridium beijerinckii NCIMB 8052 using single-nucleotide resolution RNA-Seq. BMC Genom. 2012;13: 102. doi:10.1186/1471-2164-13-102

4. Seo S-O, Janssen H, Magis A, Wang Y, Lu T, Price ND, et al. Genomic, transcriptional, and phenotypic analysis of the glucose derepressed Clostridium beijerinckii mutant exhibiting acid crash phenotype. Biotechnol J. Wiley-Blackwell; 2017;12: 1700182. doi:10.1002/biot.201700182

5. Shi Z, Blaschek HP. Transcriptional analysis of Clostridium beijerinckii NCIMB 8052 and the hyper-butanol-producing mutant BA101 during the shift from acidogenesis to solventogenesis. Appl Environ Microbiol. American Society for Microbiology (ASM); 2008;74: 7709–7714. doi:10.1128/aem.01948-08

---

## [Decision Letter · Decision Letter 1]

17 Oct 2019

Transcriptional analysis of amino acid, metal ion, vitamin and carbohydrate uptake in butanol-producing Clostridium beijerinckii NRRL B-598

PONE-D-19-22508R1

Dear Dr. Vasylkivska,

We are pleased to inform you that your manuscript has been judged scientifically suitable for publication and will be formally accepted for publication once it complies with all outstanding technical requirements.

With kind regards,

Zhiqiang Wen, Ph.D.

Academic Editor

PLOS ONE

Additional Editor Comments (optional):

Reviewers' comments:

Reviewer's Responses to Questions

**Comments to the Author**

1. If the authors have adequately addressed your comments raised in a previous round of review and you feel that this manuscript is now acceptable for publication, you may indicate that here to bypass the “Comments to the Author” section, enter your conflict of interest statement in the “Confidential to Editor” section, and submit your "Accept" recommendation.

Reviewer #1: (No Response)

Reviewer #2: All comments have been addressed

2. Is the manuscript technically sound, and do the data support the conclusions?

Reviewer #1: (No Response)

Reviewer #2: Yes

3. Has the statistical analysis been performed appropriately and rigorously? 

Reviewer #1: (No Response)

Reviewer #2: Yes

4. Have the authors made all data underlying the findings in their manuscript fully available?

Reviewer #1: (No Response)

Reviewer #2: Yes

5. Is the manuscript presented in an intelligible fashion and written in standard English?

Reviewer #1: (No Response)

Reviewer #2: Yes

6. Review Comments to the Author

Reviewer #1: (No Response)

Reviewer #2: (No Response)

7. PLOS authors have the option to publish the peer review history of their article (what does this mean?). If published, this will include your full peer review and any attached files.

Reviewer #1: None

Reviewer #2: No

---

## [Editor Report · Acceptance letter]

29 Oct 2019

PONE-D-19-22508R1 

Transcriptional analysis of amino acid, metal ion, vitamin and carbohydrate uptake in butanol-producing *Clostridium beijerinckii* NRRL B-598

Dear Dr. Vasylkivska:

I am pleased to inform you that your manuscript has been deemed suitable for publication in PLOS ONE. Congratulations! Your manuscript is now with our production department. 

With kind regards,

on behalf of

Dr. Zhiqiang Wen 

Academic Editor

PLOS ONE